# Human Rabies Treatment—From Palliation to Promise

**DOI:** 10.3390/v16010160

**Published:** 2024-01-22

**Authors:** Marian Lacy, Nonthapan Phasuk, Stephen J. Scholand

**Affiliations:** 1College of Medicine, University of Arizona, Tucson, AZ 85721, USA; mlacy@arizona.edu; 2School of Medicine, Walailak University, Nakhon Si Thammarat 80160, Thailand; nonthapan.ph@wu.ac.th

**Keywords:** rabies, encephalitic rabies, antiviral agents, lyssavirus

## Abstract

Rabies encephalitis has plagued humankind for thousands of years. In developed countries, access to preventive care, both pre-exposure and post-exposure, has significantly reduced the burden of suffering and disease. However, around the world, rabies remains a neglected tropical disease, largely due to uncontrolled dog rabies, and tens of thousands perish each year. Currently, the standard of care for management of rabies encephalitis is palliation. Heroic attempts to treat human rabies patients over the last few decades have yielded glimpses into our understanding of pathophysiology, opening the door to the development of new antiviral therapies and modalities of treatment. Researchers continue to investigate new compounds and approaches to therapy, yet there remain real challenges given the complexity of the disease. We explore and review some of the promising therapies on the horizon in pursuit of a salvage treatment for rabies.

## 1. Introduction

Rabies encephalitis is a devastating disease with nearly universal mortality. Around the world, tens of thousands die each year with little hope for survival [1]. Palliative care makes up the mainstay of treatment if it is available [2]. Many patients receive no treatment at all and die at home, unrecognized by local or international medical systems. Thus, the true burden of rabies suffering and disease is incompletely characterized. Figure 1 shows an unfortunate pediatric patient suffering from rabies.

Rabies encephalitis treatment has proven challenging for millennia. Ancient remedies included poultices, herbal preparations, or dog hair applied into the wound or used in a concoction. Some practitioners used “madstones” or animal horn (“tandok”) to counteract the poison of the biting animal [3]. Some patients in Roman times were forced into pools of water in an attempt to overcome hydrophobia [4]. Celsus, in the first century, described wound cautery as a treatment after a rabid dog bite [3]; today, this intervention would be described as post-exposure prophylaxis (PEP). Since the revolutionary breakthrough of a rabies vaccine by Dr. Louis Pasteur in the late 19th century, PEP has proved effective when instituted correctly and in a timely fashion [5]. However, no therapy has consistently demonstrated efficacy in treating disease once symptoms develop.

## 2. Challenges in Developing Effective Therapies

Limited global will and resources are barriers to developing effective therapies. This partly stems from the efficacy of PEP, but also the rarity of clinical disease in developed countries with a consequent lack of funding for research [6,7].

The unique characteristics of rabies virus (RABV) have also thwarted advancements in therapy. Rabies viruses themselves are genetically diverse with varying phenotypes. In the United States, for example, several RABV variants are in circulation—including a raccoon variant, several skunk variants, and numerous bat variants [8]. More than a dozen other lyssaviruses can cause a fatal encephalitis (i.e., clinical rabies) in humans. As new lyssaviruses are discovered, there are increasing concerns that existing immunoglobulins and vaccines may not be effective in prevention given the wide antigenic diversity [9].

The issue, then, is that no effective treatment for rabies encephalitis is available if one of the diverse variants escapes the standard modalities of prevention leading to disease or if the opportunity for post-exposure prophylaxis is unfortunately missed and the patient now has a fulminant encephalitis.

Perhaps the most formidable challenge in developing an effective rabies encephalitis treatment is our limited understanding of RABV pathogenesis and its mechanism of neuronal injury. Indeed, the study of rabies encephalitis with an eye towards treatment is complicated. Laboratory models of rabies use attenuated viral strains rather than “street” RABV strains. Compared to street viruses, attenuated viruses exhibit differences in electroencephalogram (EEG) results, rapid-eye-movement (REM) sleep, and progression to clinical disease in animal models [10]. Any laboratory-developed advances in therapy may therefore face barriers in demonstrating efficacy in natural disease.

In the clinical setting with patients, a number of factors also confound efforts for treatment. Apart from ethical considerations (discussed below), intrinsic difficulties in treatment paradigms arise. As with many central nervous system (CNS) infections, the blood–brain barrier and spinal cord barrier complicate drug delivery [7]. For example, antivirals active against RNA viruses, such as ribavirin and favipiravir, fail to reach in vivo CNS levels high enough to be efficacious against rabies [11]. In addition, RABV and other lyssaviruses attack by stealth, taking advantage of the limited immune surveillance of the central nervous system. Viral infection seems to elicit very little systemic inflammatory response initially, evading the host innate immune system. Minimal systemic inflammation preserves the blood–brain barrier, protecting the virus from host neutralizing antibodies as well as potential therapies [12].

## 3. Ethical Considerations

There are several ethical considerations in developing new therapies for rabies. Cost and healthcare resource utilization are significant. Offering pre-exposure prophylaxis (PrEP) to at-risk groups and vaccinating dogs are nearly 100% effective. Cost analysis modeling in India [13] and the Philippines [14] have estimated that large-scale PrEP in areas with a dog bite incidence of 2–5% is cost effective. Research and development of new rabies therapies may only be available to a select few patients in resource-rich countries, at least initially. Many argue that resources would be better spent on PEP and PrEP, which have proven efficacy and benefit those at highest risk in limited-resource settings.

Additionally, statistically rigorous, placebo-controlled clinical trials are impossible in an almost invariably fatal infection. In the face of such a dismal prognosis, infected patients and their families are often eager to submit to any investigational treatment, even if it has a low likelihood of success. The process of informed consent is difficult to apply in a situation in which forgoing treatment means certain death but treatment itself will almost certainly be futile. Since the burden of rabies largely falls on children in resource-limited countries, the prospect of research in children carries with it additional concerns, particularly regarding autonomy and informed consent. In infections with a high fatality rate, assignment to the placebo group could be seen as a “death sentence.” Similar challenges arose with the Ebola outbreaks in West Africa in 2014 and HIV/AIDS in the early 1980s [15]. On this basis, seven core scientific and ethical requirements have been suggested to guide clinical research during outbreaks or epidemics such as rabies with high fatality rates: (1) scientific and social value, (2) respect for persons, (3) community engagement, (4) respect for participant welfare and interests, (5) a favorable risk–benefit balance, (6) justice in the distribution of benefits and burdens, and (7) post-trial access to the tested agents that have been proven effective [16]. Admittedly, post-trial access is a crucial requirement for research in most infectious diseases, but its applicability may be limited in rabies due to the highly fatal nature of the disease. Alternative ethical considerations might include prioritizing access to established rabies prophylaxis measures like PEP for participants’ communities.

## 4. Previous and Current Therapies

There are very few documented cases of survival from rabies [17,18], and in many of those cases, survivors were left with severe permanent neurological deficits [19]. RABV and related lyssaviruses are primarily neurotropic [9], and the disruption of CNS function is catastrophic. Management therefore remains extremely difficult. Clinical features of rabies may include a panoply of neurologic symptoms—including the well-known hydrophobia and hypersalivation [20]. Brain death consistently occurs prior to cardiac arrest and failure of vegetative functions [21]. Autonomic instability, arrhythmias from the disruption of cardiac circuits, and respiratory paralysis can occur [22]. Past attempts have focused on cardiorespiratory support while attempting to preserve and restore brain function. However, this approach has proved challenging, even in more common mechanisms of brain injury, such as stroke and trauma [23].

Potential therapeutic agents for rabies have been reviewed in the past—see Table 1 [7,11,12,24,25]. A major challenge has been the failure of antivirals to reach sufficient in vivo levels in the CNS for efficacy against RABV. Interferon alpha (IFN-α), for example, held promise in early studies as a potential rabies treatment due to its broad antiviral activity. Studies in cell cultures and animal models showed some promise in inhibiting RABV replication; however, clinical trials in humans failed—even with high doses [7,26]. Several factors contributed to this—including limited ability to impede RABV spread in the CNS, debilitating side effects, and issues with tolerability. It was also crippled by timing, as IFN-α is most effective early in the disease, an Achilles heel in therapy due to the typically late-stage presentation. So, while IFN-α played an historical role in rabies treatment efforts, its limitations and lack of clear clinical benefit led to its abandonment as a viable treatment [11,27]. Similarly, ribavirin, a potent inhibitor of rabies virus (RABV) replication in vitro, lacked any clinical efficacy and therefore really has no role in rabies treatment. Ribavirin derivatives, however, have shown some potential in vitro [24].

Past authors have also noted that amantadine, minocycline, and ketamine are probably ineffective as well [7,28]. Corticosteroids and immunosuppressive agents are not recommended—particularly as they could impede viral clearance and accelerate symptom onset and death [29].

**Table 1 viruses-16-00160-t001:** Historical RABV treatment agents—failure in clinical use.

Agent	Target/Mode of Action	Notes (Effectiveness)	References
Interferons (IFN-α, b)	Upregulation of immune function	Encouraging in vitro data; failed in clinical use	[7]
Ribavirin	Nucleoside analog	Interference with RNA metabolism; required for viral replication; failed in clinical use ^1^	[25]
Amantadine	Uncertain: antiviral effect	Used in MP ^2^	[7,26]
Ketamine	Uncertain: antiviral effect	Used in MP ^2^	[7,26]
Corticosteroids	Immune regulation	Worsened disease progression	[7]

^1^ Used in the “original” Milwaukee protocol, although not used in the latest version [30]. ^2^ MP—Milwaukee protocol.

### The Milwaukee Protocol and Derivatives

Some rare but intriguing survival cases offer glimmers of hope for an effective therapy. Almost three dozen cases are reported in the literature, with almost all of them having received partial vaccination or immunizing intervention [17,18,19,31]. Gode and colleagues reported prolonged survival (>2–4 days) with good intensive care in India [32]. While none of these patients ultimately survived, these reports raised the possibility that intensive care could improve outcomes. In 2004, the sensational survival of a teenage girl who received the Milwaukee protocol (MP) was reported, raising hopes (mostly in the developed world) where intensive care could be offered [31]. The original MP involved the induction of a therapeutic coma with a cocktail of anti-excitatory agents and antiviral therapy. The protocol included ketamine, midazolam, phenobarbital, amantadine, and ribavirin. The index patient survived, albeit with some neurological sequelae. Subsequently, the MP was modified in further iterations; for example, ribavirin is no longer included [30]. However, its initial success has not been effectively reproduced. The induction of a therapeutic coma has been attempted in other patients and has been unsuccessful, with at least 53 documented failures [25]. The Recife protocol, first implemented in Brazil, is derived from the Milwaukee protocol but similarly failed to demonstrate a survival benefit [33]. Various adaptations and revisions will likely continue.

While rabies is nearly always fatal once symptoms develop, in rare situations, pursuit of an aggressive care approach might not be unreasonable. Patients have a higher likelihood of recovery if they have been vaccinated or partially vaccinated, if they have detectable rabies antibody in serum or CSF within the first week of illness, and/or if they were infected by an American bat RABV [27]. The presence of neutralizing anti-RABV antibodies at time of presentation contributes to accelerated viral clearance and increases the likelihood of survival [29]. Other relatively favorable factors include young age, lack of comorbidities, presentation with early clinical disease (such as local sensory symptoms rather than encephalopathy), and/or the absence of detectable RABV antigen and RNA in CSF. In these situations, efforts at curative therapy may be considered after informed discussion with family and acknowledgement that the outcome could result in severe neurological deficits [26]. In these clinical situations, supportive care in a critical care unit is required, often with intravenous fluid hydration, electrolyte management, antiarrhythmic drugs, antiepileptic drugs, and mechanical ventilation. Unfortunately, clinical courses are often complicated by respiratory failure, cardiac arrhythmias, autonomic instability, seizures, and/or multiorgan failure.

Thus, most rabies experts do not recommend such expensive and likely futile efforts. For now, the most accepted treatment of rabies encephalitis is palliation [29,34]. Several measures can ameliorate the suffering of patients. Optimization of the clinical setting with calm, quiet conditions that prevent unnecessary stimulation should be provided. Directed symptom management has a role as well: Due to hydrophobia, patients may suffer from severe dehydration. Oral care can be provided and patients offered foods with high water content (such as fruits). Intravenous fluid hydration may not improve the length or quality of life but could lessen the suffering of patients. Fever can be treated with antipyretics such as acetaminophen, paracetamol, ibuprofen, or aspirin. Rectal administration provides a useful route of delivery if needed. Anxiety, agitation, and seizures can be treated with benzodiazepines. Haloperidol or chlorpromazine can also manage anxiety and agitation. If patients are having difficulty managing secretions due to hypersalivation and uncoordinated swallowing, anticholinergics such as hyoscine (scopolamine) or glycopyrrolate can be used. Opioids, including morphine and fentanyl, can be used for pain.

In the Philippines, a “Starfish Palliative Care treatment protocol” was developed incorporating many of these treatment elements [34]. The authors surprisingly reported that all of the patients who received haloperidol (*n* = 22) had considerable symptom control or improvement with regards to agitation and anxiety at relatively low doses (10 to 20 mg in 24 h). There was no improvement in symptoms of hydrophobia, aerophobia, or difficulty swallowing. Benzodiazepines with or without diphenhydramine were not as effective, especially with regard to restlessness and agitation. The authors observed that “it was then possible to administer physical and personal care to the patient (which was previously unthinkable for a delirious, aggressive patient) as well as allowing emotional, social and spiritual needs to be addressed” [34]. Diphenhydramine administration improved hypersalivation and likely contributed some sedative effect. The authors noted that intramuscular routes of administration were preferred, as intravenous administration was problematic for the clinicians.

We believe that the Starfish treatment model should be prioritized for rabies palliative treatment. Table 2 outlines these essentials.

Finally, support from family members should be allowed if it provides comfort. The use of personal protective equipment (PPE) has a very important role in family–patient interaction, allowing bedside visitation. Adjunctive immunological prophylaxis (PEP) for family members would be beneficial but may be affected by costs, rabies immunoglobulin (RIG) and vaccine availability, and timing—considering the time needed for active immunity to develop following vaccination if no RIG is available.

## 5. Newer Discoveries

Although many of the compounds and therapeutic efforts tried in the clinical setting thus far have failed to date, on-going research continues. Table 3 lists some potential future antirabies therapeutics.

The development of an effective rabies treatment would probably take the shape of a protocol wherein a backbone of supportive care allows a framework to provide directly acting antivirals combined with neuroprotective and possibly functional neuronal therapies. If highly potent antivirals could be developed, then perhaps such a protocol could be streamlined to a more adaptable regimen useful in the developing world. Advances in our understanding of RABV pathogenesis could lead to pathways for drug development, much like with HIV-1 infection and the amazing progress made in that field. Of course, rabies research lacks the enormous funding and coordinated intellectual efforts that benefitted HIV-1 drug development.

One agent—favipiravir, also known as T-705—remains as a potential antiviral candidate for the treatment of rabies, albeit with reservations [40]. It is a purine nucleic acid analog (6-fluoro-3-hydroxy-2-pyrazinecarboxamide) that inhibits a wide range of RNA viruses, including RABV. It works as an inhibitor of RNA-dependent RNA polymerase (RdRp) both in vitro and in vivo. Past studies showed in an animal model that favipiravir suppressed RABV replication and significantly prolonged the survival period in RABV-infected mice [41]. However, it should be noted that the survival rate was not improved. Problems with drug delivery to the CNS—i.e., the blood–brain barrier (BBB) and blood–spinal cord barrier (BSCB) might be overcome with different drug delivery systems into the CNS.

Other agents in development have shown some promise, including oligonucleotides, small molecules, proteins, and immunotherapy. Short interfering RNAs (siRNAs) are short nucleotide sequences (about 21–23 nucleotides in length) that silence complementary mRNA in the cell, thereby inhibiting viral replication in vivo [35,36]. However, a “match” in the sequences is required, which may prove difficult given the diversity of RABV strains. A recent study showed the significant inhibition of RABV in a post-infection model using siRNA [42]. Other approaches with RNA attempt to replicate the body’s innate cellular defense mechanisms: artificial microRNAs (amiRNAs) in some experiments have effectively inhibited viral replication in vivo [43]. Unfortunately, this line of research has not advanced much in the last decade.

Small molecules and proteins may play a role in treatment [37,44]. One study involving an ATP-binding cassette family E1 (ABCE1), which is a host protein involved in ribosome recycling, showed marked antiviral potency and an excellent therapeutic index in cell culture [45].

A compound known as TMP269, which is a small molecular inhibitor of class IIa histone deacetylase (or HDAC), was found to significantly inhibit RABV replication [37] in in vitro experiments. It showed a dose-dependent benefit with reduced RABV viral titers and protein levels at an early stage in the viral life cycle. TMP269 appears to affect cellular processes—reducing the innate inflammatory immune response, as well downregulating autophagy. Autophagy is thought to enhance RABV replication, so that TMP269 inhibition of autophagy leads to a decrease in RABV replication.

Another compound, bardoxolone methyl (CDDO-Me), was seen to have an antiviral effect against RABV in in vitro experiments [38]. CDDO-Me works as an antioxidant inflammatory modulator and is a potent nuclear factor erythroid-derived 2-like 2 (Nrf2) activator. Nrf2 is thought to help protect against virus-induced injury and inflammation. Experiments showed that CDDO-Me suppressed viral growth via Nrf2-dependent signaling and inhibited RABV infection to a degree comparable to that of ribavirin and significantly better than that of favipiravir (T-705) at lower doses.

Indeed, studies continue to elucidate further mechanisms of host regulation in viral replication with regards to RABV. Zhang B et al. demonstrated an important role of the tripartite-motif protein (TRIM) family of proteins affecting RABV replication [46,47]. Specifically, they showed that TRIM21 regulated the secretion of type I interferon during RABV infection and suggested therefore that TRIM21 was a potential target for rabies treatment and/or management. They also proposed that TRIM25 might be a possible target given its regulatory role, although more work is needed. Further research and understanding of the mechanisms of neuronal injury and dysfunction in rabies may open the door to novel neuroprotective therapies and therapeutic targets [48]. Cannabinoids are one class of compounds that may provide a useful role in neuroprotection. Cannabinoid receptors are found throughout the CNS and peripheral nerves, as well as in the microglia and astrocytes of the brain. Cannabis-related drugs are thought to regulate cell homeostasis and facilitate a cytoprotective effect, promoting viral clearance [29].

These and other useful insights of the host–pathogen interaction create a good foundation for future discoveries [49].

### 5.1. Future Directions

The future of rabies encephalitis treatment is not entirely bleak.

Monoclonal antibodies (mAbs) were developed for rabies decades ago as a means to augment prophylaxis by neutralizing the virus in the periphery before invasion and CNS infection [50]. Newer research has shown that there could be useful therapeutic efficacy for some mAbs in established CNS infection. One study evaluated the therapeutic efficacy of F11, an anti-lyssavirus human monoclonal antibody, and showed that a single dose limited viral load in the brain and reversed disease signs following infection with a lethal dose of lyssavirus [39].

Another group [51] experimented with two human neutralizing monoclonal antibodies, RVC20 and RVC58, shown to be effective in treating symptomatic rabies. Their study showed that the mAbs recognize and lock the RABV-G protein in its pre-fusion conformation and display multiple mechanisms of action which could help the development of these mAbs as treatment for human rabies.

In the future, the therapeutic delivery of immunotherapy might be administered in novel ways as well. One group of researchers created a construct (named RABV-62scFv) consisting of a live-attenuated recombinant RABV expressing a highly RABV-neutralizing scFv antibody (62-71-3) linked to the fluorescent marker mCherry [52]. This approach presumably would deliver the “drug” where needed most by essentially hijacking the RABV pathogenic process. Experiments showed encouraging scFv production and subsequent virus neutralization.

Finally, other future directions of therapy include the development of immunostimulatory substances, such as oligodeoxynucleotides, that have been shown to be effective adjuvants for rabies vaccines and may potentially be useful for therapy of rabies [45]. Years of research and development are likely needed before any practical applications emerge.

### 5.2. One Medicine and the Development of Dog-Based Research Models

Future investigations regarding rabies treatment could certainly benefit from more robust research efforts, particularly on the clinical side. Knobel and colleagues proposed an approach to accelerate the development of an effective therapy for rabies through the study of naturally infected dogs [53]. They outlined the development of a model utilizing critical care with investigational treatment and supportive therapy that could translate into significant benefits also on the human side. This could facilitate the study of combination therapies more rigorously to address disease processes—since currently, there are no organized human trials for treatment. In addition, dog-based research may help identify biomarkers for prognosis and therapeutic response, as currently there are none. The authors went on to introduce the “Canine Rabies Treatment Initiative, a non-profit organization with the mission to apply a One Medicine approach to the investigation of diagnostic, prognostic, and therapeutic options for rabies in naturally infected dogs, with the goal of transforming rabies into a treatable disease for all patients” [54].

Hopefully, this fresh approach will lead to sorely needed breakthroughs for rabies patients.

## 6. Conclusions

Rabies encephalitis remains a neglected tropical disease—disproportionately affecting the world’s most vulnerable populations—and continues to present formidable clinical challenges. Valiant salvage efforts have improved our understanding of RABV pathology and natural history. Continued efforts in the rabies research community have provided further insights for potential treatments. Newer approaches to treatment might best include a multi-pronged approach with antiviral therapy, immunotherapy, and neuroprotective therapy.

A renewed attention to rabies treatment is needed to restimulate efforts to develop effective therapies. Education and outreach such as the “Zero by 30” campaign supported by the World Health Organization and the Global Alliance for Rabies Control can play a major role by encouraging global collaboration and support [55]. For now, the prevention of rabies remains tantamount, as promise for a truly effective treatment remains just out of reach.

## Figures and Tables

**Figure 1 viruses-16-00160-f001:**
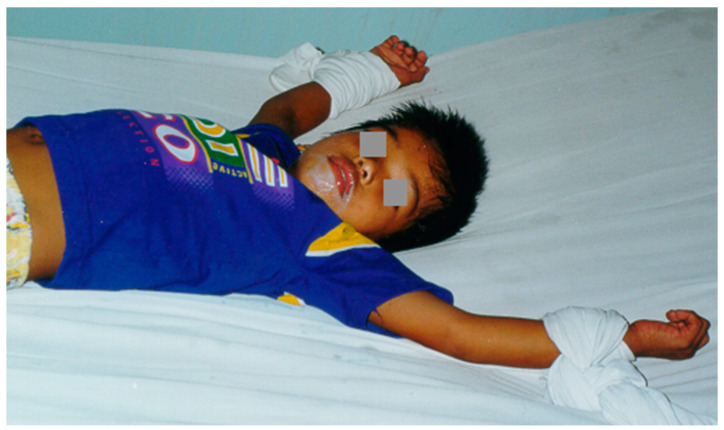
This young child is suffering from rabies encephalitis–note the thick salivary secretions, teeming with rabies virus. Restraints are often used in clinical practice because of symptoms of agitation.

**Table 2 viruses-16-00160-t002:** Essentials of the Starfish Palliative Care treatment protocol for rabies patients.

Symptom(s)	Medication/Route of Administration	Adult Dosage	Pediatric Dosage
	Paracetamol PR	1000 mg q 4–6 h	125–500 mg q 4–6 h
Fever	Ibuprofen PR	300–400 mg q 6–8 h	N/A
	Aspirin PR	450–900 mg q 4–6 h	N/A
Agitation,	Haloperidol IM, SC	5 mg q 1 h to effect ^1^, then q 4–6 h	See note ^2^
Anxiety, and	Chlorpromazine IM	25–50 mg q 6–8 h	0.5 mg/kg q 6–8 h
Restlessness	Chlorpromazine PR	100 mg q 6–8 h	
	Diazepam IM	20 mg q 2 h	0.1–0.3 mg/kg q 1–4 h
	Diazepam PR	10 mg q 1–4 h	
	Lorazepam IM	2 mg q 4 h	N/A
Sedation and	Diphenhydramine IM	50–100 mg q 4–6 h (max 400)	5 mg/kg/24 h (max 300)
Secretion management			

^1^ Given every hour × 3 for a loading dose in the original study [34]. ^2^ Age 1 month–12 y: haloperidol 25–85 microg/kg/24 h; 12–18 y: 1.5–5 mg/24 h.

**Table 3 viruses-16-00160-t003:** Potential RABV treatment agents—promise for the future.

Agent	Target/Mode of Action	Notes (Effectiveness)	References
Oligonucleotides ^1^	Interference with RNA	Targets can be designed	[35,36]
TMP269	Affects cellular processes		[37]
Bardoxolone methyl (CDDO-Me)	Affects cellular processes		[38]
Monoclonal antibodies (mAbs)	Antiviral effect	(Usage beyond prevention)	[39]
Favirapivir	Purine nucleic acid analog	Inhibits RNA pol; mixed results	[40,41]
Cannabinoids	Neuroprotection	An emerging area of research	[29]

^1^ Would include antisense oligonucleotides (ASOs), which bind to complementary sequences of viral RNA, plus small interfering RNA (siRNA) molecules.

## Data Availability

Published literature.

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
