# Peer review of "Human Rabies Treatment—From Palliation to Promise"

_viruses, 2024, doi:10.3390/v16010160_

Round 1

Reviewer 1 Report

Comments and Suggestions for Authors

This is a well, clear written paper describing focused on continued efforts in the rabies research for potential treatment of the disease. Description of new approaches to treatment including a multi-pronged approach with anti-viral therapy, immunotherapy and neuroprotective therapy. This renewed attention to rabies treatment may stimulate efforts to develop effective therapies. New potential therapeutic agents for rabies have been also reviewed.

Author Response

Thank you very much for this kind review. We are grateful for your time and effort. It means alot!  I hope you enjoyed reading about some of the treatment advances in rabies research - although admittedly we still have a long way to go!  Thanks so much.

Stephen Scholand

Reviewer 2 Report

Comments and Suggestions for Authors

A comprehensive review article about treatment of human rabies.

1Lines 11 – 13 – It is not clear that attempts to treat human rabies patients has advanced our understanding of the pathogenesis of rabies at all.

2Line 90 – Item #7 does not apply in rabies.

3Line 93 – Ref #17 is poor citation

4Lines 98/99 – An absence of neuronal apoptosis at autopsy has been reported. The statement is incorrect.

5Lines 115/116 – The report on ribavirin derivatives is in vitro (according to the title of the citation).

TTable 1 – Ribavirin was also used in the Milwaukee protocol.

7Lines 126/127 – Almost three dozen instead of about a dozen.

8Line 127 – Actually another case from Mexico did not receive rabies vaccine prior to the onset of clinical rabies.

9Line 138 – Ref #32 is a wrong citation.

L Lines 140/141 – delete “until a truly effective rabies encephalitis treatment is developed.”

1Line 145 – Ref #26 seems incorrect.

L Lines 168/169 – More information should be presented on the “Starfish protocol.”

1 Line 171 – Timing is problematic with immunological prophylaxis.

1Lines 183/184 – No need to mention the Milwaukee protocol here.

1Lines 194-196 – It should be added that the survival rate was not improved.

1There is no mention of evaluating potential therapies in naturally infected dogs – see publication of D Knobel.

1Is there any evidence that WHO and GARC have any serious interest in the treatment of human rabies? They seem preoccupied with the great challenge of preventing the disease, which has much greater impact on human lives.

  Minor specific points:

1Line 39 – research rather than clinical trials

1Line 44 – delete “new RABV variants and”

2Line 49 – postexposure prophylaxis instead of immunotherapy

2Line 49 – delete “declining”

Author Response

Thank you so much for your incredible, wonderful and detailed review of our paper. We were very grateful for the suggestions and points you made - and we are eternally thankful for making this paper so much better.  Your review really made a huge difference - particularly with regards to the Starfish protocol and the dog based One Health research proposal.  Attached we have a point-by point response to each of your queries. THANK YOU again - this was just amazing!  Stephen Scholand and co-authors

Reviewer 3 Report

Comments and Suggestions for Authors

viruses-2821807

This brief review, by Lacy et al., of past-present-and future anti-rabies virus therapies and treatment modalities was extremely well-written, succinct, inclusive, up-to-date and timely. Great summaries and topic flow.  All major issues touched on; including human ethical considerations, resources and availabilities in lesser developed nations, emergency use therapeutic protocols that have been or continue to be explored, and the latest antiviral testing in vitro and in vivo (animal models).  Enjoyed reading it.

Extremely minor: page 2, line 62 is there a missing citation with ‘11’ favipiravir not discussed in that citation, the later ref 35 discusses some positive results that need pursing, although this is reviewed later.

Author Response

Thanks so much for your wonderful review of our paper. We are grateful for all the time and effort you took to help us.  Someday I hope we can repay the favor.  Warmest regards - Stephen Scholand and co-authors